# Single-Cell Transcriptomic Profiling of the Mouse Testicular Germ Cells Reveals Important Role of Phosphorylated GRTH/DDX25 in Round Spermatid Differentiation and Acrosome Biogenesis during Spermiogenesis

**DOI:** 10.3390/ijms24043127

**Published:** 2023-02-04

**Authors:** Raghuveer Kavarthapu, Rajakumar Anbazhagan, Soumitra Pal, Maria L. Dufau

**Affiliations:** 1Section on Molecular Endocrinology, Division of Developmental Biology, *Eunice Kennedy Shriver* National Institute of Child Health and Human Development, National Institutes of Health, Bethesda, MD 20892, USA; 2Neurobiology-Neurodegeneration and Repair Laboratory, National Eye Institute, National Institutes of Health, Bethesda, MD 20892, USA

**Keywords:** GRTH/DDX25, round spermatid differentiation, single-cell RNA-seq, spermiogenesis, acrosome

## Abstract

Gonadotropin-regulated testicular RNA helicase (GRTH)/DDX25 is a member of DEAD-box family of RNA helicase essential for the completion of spermatogenesis and male fertility, as evident from GRTH-knockout (KO) mice. In germ cells of male mice, there are two species of GRTH, a 56 kDa non-phosphorylated form and 61 kDa phosphorylated form (pGRTH). GRTH Knock-In (KI) mice with R242H mutation abolished pGRTH and its absence leads to infertility. To understand the role of the GRTH in germ cell development at different stages during spermatogenesis, we performed single-cell RNA-seq analysis of testicular cells from adult WT, KO and KI mice and studied the dynamic changes in gene expression. Pseudotime analysis revealed a continuous developmental trajectory of germ cells from spermatogonia to elongated spermatids in WT mice, while in both KO and KI mice the trajectory was halted at round spermatid stage indicating incomplete spermatogenesis process. The transcriptional profiles of KO and KI mice were significantly altered during round spermatid development. Genes involved in spermatid differentiation, translation process and acrosome vesicle formation were significantly downregulated in the round spermatids of KO and KI mice. Ultrastructure of round spermatids of KO and KI mice revealed several abnormalities in acrosome formation that includes failure of pro-acrosome vesicles to fuse to form a single acrosome vesicle, and fragmentation of acrosome structure. Our findings highlight the crucial role of pGRTH in differentiation of round spermatids into elongated spermatids, acrosome biogenesis and its structural integrity.

## 1. Introduction

Mammalian spermatogenesis is a complex and dynamic process of germ cell proliferation and differentiation within the seminiferous epithelium of the testis to facilitate the production of sperm [1,2]. Male fertility relies upon steady-state spermatogenesis, beginning with self-renewing spermatogonial stem cells, giving rise to progenitor spermatogonia that initiate spermatogenic differentiation to ultimately form mature spermatozoa [3,4]. The germ cells undergo precise transitions between multiple cell types and cellular processes, including mitosis, meiosis and spermiogenesis, which is accompanied by spermatid elongation and chromatin repackaging [3,4]. The process of germ cell development from spermatogonia into mature sperm depends on the integrated expression of a range of genes in a precise temporal order [5,6]. In the adult mammalian testis, the majority of mRNAs are associated with specific RNA binding proteins as a messenger ribonuclear protein (mRNP) complex. mRNAs are transported from nucleus to the cytoplasm where messages are translationally repressed, presumably in the Chromatoid Bodies (CBs) of round spermatids (RSs). Translational activation of stored mRNAs transported to polyribosomes at later times during spermatid development is essential for the progression of spermatogenesis [7,8].

Gonadotropin-regulated testicular RNA helicase (GRTH), also known as DDX25, is a novel testis-specific RNA helicase identified in our laboratory, which plays an essential role in the progression of spermatogenesis [9,10,11,12,13,14]. GRTH belongs to DEAD-box protein family of RNA helicases explicitly expressed in the Leydig cells and germ cells of testis and is transcriptionally regulated by gonadotropins via androgen [10,11,12]. It is involved in multiple functions as a post-transcriptional regulator of specific genes in germ cells during spermatogenesis. GRTH is an integral constituent in the mRNP complex that transports specific germ cell mRNAs from the nucleus to cytoplasmic sites including CBs for storage prior to their translation during spermatogenesis [12,13,14]. CBs are non-membranous organelles residing in the cytoplasm of RSs where specific mRNAs are stored and later processed during spermiogenesis [8,11,12]. We have shown that pGRTH binds to actively translating polyribosomes and might play an important role in translation of specific genes in germ cells, such as transition proteins (*Tnp1*/*Tnp2*), protamines (*Prm1*/*Prm2*), phosphoglycerate kinase and testicular angiotensin-converting enzyme [15,16]. We previously reported two forms of GRTH protein in germ cells, a 61 kDa phosphorylated species exclusively found in the cytoplasm and a 56 kDa non-phosphorylated species in the nucleus, also present in the cytoplasm [12,13,14]. The 61 kDa phosphorylated GRTH (pGRTH) protein in the cytoplasm participates in shuttling of specific mRNAs in and out of the CBs, and associates with polyribosomes for translation. While the 56 kDa non-pGRTH protein is involved in the export of mRNAs from nucleus to cytoplasm [11,14]. GRTH knock-out (KO) mice are infertile with complete loss of elongating spermatids (ESs) and sperm due spermatogenic arrest at step 8 of RS development [11]. In the RSs of these null mice, the size of the CBs was markedly reduced, indicating either a decrease in or loss of mRNAs essential for progression of spermiogenesis [11,13]. Our previous studies identified a missense mutation of Arg to His at position 242 (R242H) in the GRTH gene in 5.8% of Japanese men with non-obstructive azoospermia [17]. In vitro experiments performed by overexpressing the human mutant GRTH (R242H) construct in COS-1 cells revealed the loss of cytoplasmic 61 kDa pGRTH species, while maintaining the expression of a 56 kDa non-phospho form [17]. We established that GRTH was phosphorylated by Protein Kinase A (PKA) at the T239 residue. cAMP treatments increase the phosphorylation of GRTH in COS-1 cells by regulating PKA activity [18]. Mutations in E165/K240/D237 amino acids surrounding the R242 residue at the GRTH–PKA interface results in marked reduction in GRTH phosphorylation [18]. Subsequently, we created transgenic GRTH Knock-In (KI) mice bearing the human GRTH gene with the R242H mutation, which lacks the 61 kDa cytoplasmic pGRTH form but retains the non-pGRTH form in the nucleus [19]. The homozygous KI mice are infertile with the absence of mature sperm due to failure of RSs to elongate while showing normal mating behavior. In these KI mice, loss of pGRTH has significant effects on the levels of mRNA and protein of TP2, PRM2 and TSSK6 [19]. In the present study, we performed single-cell RNA-seq (scRNA-seq) profiling of testicular germ cells isolated from KO, KI and WT mice using the 10× Genomics Chromium platform to understand the role of phospho and non-phospho GRTH in gene expression dynamics in different stages of germ cell development during spermatogenesis. The scRNA-seq data from different cell types (meiotic to haploid germ cells) will provide a detailed transcriptome landscape, which then will allow us to better understand the changes in the gene expression profiles at specific stages of germ cell development/differentiation during spermatogenesis. Using scRNA-seq transcriptomic profiles from the KO and KI germ cells, we also want to find out various biological processes affected due to the loss of pGRTH that will delineate the precise role of pGRTH in RS differentiation during spermiogenesis.

## 2. Results

### 2.1. Single-Cell RNA-Sep Establishes Different Germ Cell Clusters in the Adult Mouse Testis

Here, we performed scRNA-seq of testicular cells isolated from seminiferous tubules of WT, KO and KI mice. Standard data processing of the total ~13,500 cells showed an average of nearly 25,000 mapped reads/cell with a median of 2100 genes/cell, all of which indicate the robustness of the sequencing method (Appendix A). Unsupervised clustering of testicular cells via tSNE analysis identified 15 clusters representing the complete spermatogenic lineage, with a small contribution from somatic cells (Figure 1A). We created tSNE clustering of all the cells based on individual samples (WT, KO and KI; Figure 1B). Heatmap showing top-20 differentially expressed genes (DEGs) in all the 15 clusters (Figure 1C). Based on previously known cell-type marker gene expression, we identified clusters 13 to 15 as somatic cells and clusters 1–12 correspond to germ cells (Figure 1D). Clusters 13, 14 and 15 were identified as macrophages, Sertoli cells and myoid cells, respectively (Appendix A). Analysis of the top-10 most DEGs between cell clusters 1 and 12 identified characteristic germline-specific markers for each cell type, classified into four major germ cell types: spermatogonia (SG), spermatocytes (SPs), round spermatids (RSs) and elongated spermatids (ESs). We identified cluster 1 as SG that expresses *Dmrt1*, *Uchl1*, *Ncl*, *Hells* and *Dazl* genes (Figure 1D). Clusters 2–5 represent distinct states that are transitioning from differentiating SG to SP. These cells express early SP marker genes involved in meiotic division and RNA binding, such as *Sycp3*, *Hormad1*, *Piwil1* and other late SP-related genes *Tex101*, *Rsph1* and *Calm2* (Figure 1D). We then searched for clusters with marker genes that are very specific to spermiogenesis (differentiation of RS to ES). We identified clusters 6–8 as RSs that highly express acrosome-related genes, such as *Acrv1*, *Spaca1/3*, *Catsper3/4*, *Eqtn* and *Tssk1*. Finally, clusters 9–12 belong to ESs that specifically express genes involved in chromatin condensation/packing and elongation of spermatids, such as *Tnp1*, *Tnp2*, *Prm1*, *Prm2*, *Tssk6*, *Spem1*, *Ccdc91* and *Oaz3* (Figure 1D). Dot plot analysis showed average expression of specific marker genes used for identification of cell types (Figure 2A). We then created the tSNE clustering of all spermatogenic and somatic cells based on the markers identified for each cell type (Figure 2B).

### 2.2. Pseudotime Trajectory Analysis of Germ Cell Development during Spermatogenesis

For further downstream analysis, somatic cell clusters were excluded from gene expression analysis, and we focused on germ cell clusters 1–12. We clustered only the germ cells from WT, KO and KI samples undergoing different stages of spermatogenic development into four major cell types (Figure 3A). We performed pseudotime analysis on these major germ cell clusters, which revealed a long, continuous trajectory of germ cell development, sometimes separated by distinct states, indicating temporal order of spermatogenesis in the WT sample (Figure 3B). However, in the case of KO and KI samples, this continuous trajectory of germ cell development is halted in the RS stage, where RSs fail to elongate completely, with no distinct ES (Figure 3B).

### 2.3. Transcriptional Profiles Are Significantly Altered in RS of KI and KO Mice during Spermiogenesis

We identified specific cohorts of genes with maximal expression in clusters 6–12 where the germ cells are transitioning from early stages of RS to ES (condensed spermatid) during spermiogenesis (Figure 4A). We then examined the changes in gene expression of selected specific marker genes during the spermiogenesis process in WT, KO and KI samples. The dot plot analysis revealed significant changes/down-regulation in the transcriptional profiles of specific genes in spermatids at different stages (early to late) of development/differentiation during spermiogenesis in both KO and KI mice compared to WT (Figure 4B). In further studies, we performed differential gene expression analysis on a RS cluster to find out DEGs between KI vs. WT and KO vs. WT groups (Appendix A). Functional enrichment analysis on DEGs was performed to gain more insight on the biological functions that are significantly altered. GO analysis on DEGs for both groups revealed that changes in the biological process were significantly enriched in spermatid development/differentiation, microtubule-based movement, ribonucleoprotein complex assembly and translation process (Figure 4C,D). Changes in the cellular component for DEGs were primarily enriched in secretory granule/acrosome vesicle, microtubule and ribosome (Figure 4C,D). qRT-PCR validation of the selected genes related to acrosome function (*Tssk1*, *Acrv1*, *Iqcf1*, *Spata25*, *Eqtn*, *Capza3*, *Atp6v1E2*) and translation process (*Eif4e*, *EiF1*, *Eif2s2*, *Rps26*, *Rsp29*) showed significant down-regulation in the RS isolated from KO and KI mice (Figure 5A,B). We found cell-cycle-related genes, such as *Ccna1*, *Cdc20*, heat-shock protein gene *Dnajc3* and dynein axonemal gene *Dnah8,* were up-regulated in KO and KI mice. The qPCR results corroborate with DEGs from RS cluster in our scRNA-seq data (Figure 5A,B).

### 2.4. Morphological Changes in Acrosome Formation in Developing RS Revealed by TEM

During the spermiogenesis process, acrosome formation is a crucial event for the development/differentiation of RSs into elongating spermatids. Golgi apparatus plays a critical role in the formation of the acrosome that involves the fusion of pro-acrosome vesicles (PVs) derived from the trans-Golgi network (TGN) for the formation of a single, large acrosomal vesicle attached to the nuclear envelope. Since the acrosome-related genes were significantly affected, we looked at the morphological changes in the acrosome formation and its structural integrity during spermiogenesis. We used FITC-tagged peanut-agglutinin (PNA) staining to label the acrosome in the developing spermatids. PNA staining in WT RS in the Golgi phase accumulates homogeneously as a single distinct acrosomal vesicle (Figure 6A,B). In the cap phase, the WT RSs display a characteristic single uniform layer of acrosomal caps attached to the nucleus (Figure 6G,H). While both KO and KI spermatids display acrosomal defects, PNA staining showed a dispersed pattern with more than one acrosome vesicle in the Golgi phase (Figure 6C–F,I–L). We observed acrosomal structures appeared discontinuous, indicating that they were abnormally fragmented without the formation of a distinct cap-shaped acrosome (Figure 6C–F,I–L).

To understand the failure of acrosome formation in detail, we examined the ultrastructure of developing RSs using TEM. We observed noteworthy abnormalities/defects in the acrosome structure and Golgi complex (which play an essential role in acrosome vesicle formation) in RS of KI and KO. In WT mice, we could observe all the four phases of acrosome biogenesis distinguished by the progress of acrosome growth and the corresponding thickening of the nuclear envelope (Appendix A), while the KI and KO mice displayed only two phases (Golgi phase and Cap phase) of the acrosome formation with several abnormalities (Appendix A). TEM analysis of WT early RS in the Golgi phase revealed the formation of a single large acrosome granule from the fusion of PVs (Figure 7A,B). The acrosomal granule is bound to the nuclear envelope. While in KI and KO mice, we observed multiple acrosome granules and vesicles of different sizes due to failure of the PVs (indicated in arrows) to fuse and form a single acrosome granule in the Golgi phase (Figure 7C–F), in RS of WT mice. We observed PVs of uniform size situated between the TGN and the nuclear membrane in WT mice. In the RS of KI and KO, we noticed a greater number of PVs and they appeared irregular in size and much larger than those in the WT mice (Figure 7D,F). These observations suggest that there are defects in the budding of vesicles from the TGN in KI and KO spermatids. Moreover, in KI and KO RS, we frequently noticed that the Golgi apparatus showed a considerable disorganization and fragmentation with a decreased proportion of flattened membrane stacks, and the lamellar structure of TGN formed loose spirals (Figure 7F,J,L). As spermiogenesis proceeded to the cap phase, acrosomal abnormalities in RS became more obvious. In the cap phase of WT mice, the acrosomal granule becomes enlarged, begins to flatten upon touching the nuclear envelope and spreads over the nucleus to form a cap-like structure (Figure 7G,H). In contrast, the RS of KI and KO mice displayed relatively large vesicles that were often seen adjacent to incompletely formed or developing acrosome structures that are fragmented at different regions (Figure 7I–L). We also observed other morphological defects, such as vacuolated acrosome, acrosome structure not fully attached to the nuclear envelope (Figure 7D,L) and, in some cases, complete loss of acrosome. These studies clearly indicate the role of GRTH in the formation of acrosome granules and maintenance of the structural integrity of the acrosome.

## 3. Discussion

Mammalian spermatogenesis is a complex developmental process controlled by the testis niche/somatic cells that involves differentiation of adult spermatogonial stem cells into mature spermatozoa. It requires intricate interactions between the germ cell and somatic cells. A seminiferous tubule’s cross section contains several types of somatic and germ cells. It has been challenging to profile different cell types at various developmental stages due to this cellular heterogeneity. scRNA-seq profiling can comprehensively define the transcriptomes of a cell lineage and can also delineate the degree of cellular heterogeneity, which bulk RNA sequencing is unable to accomplish [20]. scRNA-seq provides an invaluable tool for studying the role of GRTH in individual germ cells at different stages of spermatogenesis simultaneously at very high resolution in GRTH transgenic mice models [6,21]. In this study, we utilized the scRNA-seq method from 10× genomics for profiling germ cell populations, which rapidly progress through many developmental transitions. We created single-cell transcriptomic profiles of testicular germ cells isolated from WT, KO and KI mice to understand the gene expression dynamics during spermatogenesis.

We used previously well-defined stage-specific markers to identify somatic and spermatogenic cell types in the 15 clusters generated by cell partitioning via tSNE analysis [6,21,22]. Clusters 1–12 displayed cell-specific markers for germ cells undergoing different stages of development, which were classified into four major cell types as SG, SP, RS and ES. The unsupervised ordering of germ cells allowed us to reconstruct a complete developmental trajectory of the spermatogenic process in transgenic mice models to determine the precise role of pGRTH. Pseudotime and clustering analyses reveal dynamic gene expression patterns during spermatogenesis. Notably, the germ cell clusters 1–12 in the WT sample formed a wave-like continuum that replicated the temporal order of spermatogenesis and was occasionally divided by discrete bottlenecks. However, this continuous trajectory of germ cells during spermatogenesis in KO and KI mice is halted at the RS stage where elongation takes place. This incomplete trajectory of the spermatogenic process is consistent with the morphological data from KO and KI mice where the seminiferous tubules totally lack ES and sperm because the RSs at step 8 fail to elongate.

By examining the transcriptome profiles of individual germ cells at various spermatogenic phases, we found that the loss of pGRTH in KI and both non-phospho and pGRTH in KO has no significant impact on the biological functions of mitotic and meiotic cells. However, during spermiogenesis, haploid cells (RS) showed significant changes in their transcriptional profiles and morphological development. Gene ontology enrichment analysis of DEGs in RS clusters of KO and KI revealed several genes related to spermatid development/differentiation, translation process and acrosome vesicle were significantly altered. Ribosomal subunit assembly genes and translation initiation factor genes were down-regulated in KO and KI mice. In this study, we found transcripts of acrosomal genes, such as *Tssk1*, *Acrv1*, *Spata25*, *Eqtn*, *Iqcf1*, *Capza3* and *Atp6ve,* were significantly down-regulated in RS of KI and KO mice during spermiogenesis. TSSK1 is a testis-specific serine kinase found in post-meiotic germ cells and localized in the acrosome of RS and ES and sperm flagellum. Targeted deletion of the *Tssk1 gene* in mice causes male infertility and morphological defects, such as failure to form ESs, apoptosis of spermatids and appearance of several degenerating RS in the lumen of epididymis [23]. Histological observations of testis of both KO and KI mice also showed very similar morphological defects found in TSSK1 KO mice [11,19]. The *Acrv1* gene encodes a male germ cell-specific acrosomal vesicle protein 1 that is first noticed in acrosomal vesicles in the Golgi phase of RS and in subsequent stages of ES. It is abundantly expressed in both the acrosomal matrix as well as in the inner acrosomal membrane and has a role in spermatogenesis [24]. There were no KO mice studies carried out on the *Acrv1* gene to understand its precise role during acrosome formation. EQTN (Equatorin), also known as acrosome formation-associated factor (AFAF), is an acrosomal membrane-anchored protein involved in the process of fertilization and in acrosome biogenesis. Eqtn-KO males have reduced fertility and sperm-egg adhesion [25]. IQCF1 (IQ Motif Containing F1) is an acrosomal protein, which plays an important role in sperm capacitation and the acrosome reaction. Iqcf1 null mice were significantly less fertile because of reduced sperm motility and the acrosome reaction [26]. CAPZA3 protein, such as IQCF1, is expressed in acrosomal regions and has a multi-functional role from acrosome biogenesis to capacitation and acrosome reaction [27]. ATP6V1E2 is a subunit of a sperm-specific V-ATPase protein that is expressed in secretory acrosomes and involved in acidification of acrosome essential for fertilization [28]. *Spata25* (also known as *Tsg23*) mRNA is predominantly expressed in SP and RS of humans and mice. *Spata25* expression was significantly decreased in patients with obstructive azoospermia and in testis of an azoospermic mouse model, suggesting its role in spermatogenesis [29]. We know from our previous studies that pGRTH is involved in maintaining the stability of germ-cell-specific mRNAs that are stored in the CBs of RS until these RNAs are ready for translation in later stages of RS development [14,19]. Therefore, loss of pGRTH results in a significant reduction in mRNA levels of the above-mentioned acrosomal genes observed in our present study, which can eventually lead to several defects in acrosome formation noticed in RS of KI and KO. We found significant reductions in transcript levels of different eukaryotic translation initiation factors (*Eif4e*, *EiF1*, *Eif2s2*) associated with small ribosomal units and essential for formation of preinitiation complex for translation process. *Rps26* and *Rps29* code for ribosomal protein S26 and S29, respectively, and are the components of 40S ribosomal subunit required for protein translation, which were also down-regulated in the RS of KO and KI. We previously reported that pGRTH binds to actively translating polyribosomes and may be involved in the translation of a few testis-specific germ cell mRNAs, such as *Tnp2* and *Prm2* [15,16]. We believe that down-regulation of these translation initiation factor genes and *Rps26/29* in GRTH transgenic mice may profoundly affect the translation process, leading to the loss of TP2 and PRM2 proteins and reduction in TSSK6 protein in the testis of KI [19]. scRNA-seq and qPCR analysis showed *Cdc20*, *Ccna1*, *Dnajc3* and *Dnah8* genes were up-regulated in RS of KO and KI. CDC20 is a cell division cycle 20 protein, which is highly expressed in both SP and RS, and it is involved in mitosis and meiosis processes. Overexpression of CDC20 in HeLa cells showed that a higher frequency of formation of multinuclear cells and in oral squamous cell carcinoma cells leads to an impairment in the spindle assembly checkpoint [30]. In our earlier studies, we observed the formation of several multinucleated giant cells in the testis sections of KI mice [19]. CCNA1 (Cyclin A1) is predominantly expressed in late pachytene SP and essential for meiotic cell division during spermatogenesis [31]. Elevated levels of CCNA1 cause chromatin condensation and apoptosis in renal, ovarian and lung carcinoma cells [32]. We noticed increased apoptosis in RS of KI and KO in our previous studies, which can be explained by increased expression of *Ccna1* mRNA noticed in our present study [11,19]. DNAJC3 (DnaJ heat-shock protein family member C3) is a molecular chaperon expressed in most of the cells and is involved in the unfolded protein response during endoplasmic reticulum (ER) stress [33]. DNAH8 is dynein axonemal heavy chain 8 protein expressed in SP and spermatids, which is involved in cell motility and sperm motility. It has been reported that DNAH8 variants lead to morphological abnormalities in the sperm flagella and male infertility [34]. The fact that these changes in gene expression are found in RS of both transgenic models clearly indicated the role of pGRTH in the expression of these genes since the non-pGRTH is still present in the KI mice.

In mammals, spermatid differentiation occurs concurrently with acrosome biogenesis during spermiogenesis, which is defined by the transition of RSs into mature sperm. The process of spermiogenesis in mice is characterized in 16 steps, with acrosome biogenesis being a key event [35]. Acrosome biogenesis is classically divided into four major phases: Golgi, cap, acrosome and maturation. During the Golgi phase, the TGN gives rise to several small proacrosomal vesicles that are required for the formation of a mature acrosome. These proacrosomal vesicles fuse to form a large single acrosomal granule that associates with the nuclear envelope [35,36]. In the cap phase, the acrosome grows larger in size and spreads over the nucleus to form a cap-like structure. The “acroplaxome,” a crucial component that aids spermiogenesis, can be located close to the distal end of the growing acrosome. In the acrosome phase, the cap-shaped acrosome elongates further, encircling the dorsal border, protruding apically and, at the end of the maturation phase, the acrosome structure is complete [35,36]. Immunofluorescence detection of acrosomes using FITC-PNA staining revealed several abnormalities throughout the acrosome development during the spermiogenesis process. When we carefully examined the ultra-structures of RSs using TEM, we found severe defects in acrosome biogenesis in the KO and KI mice from the early stages of spermatid development. Most frequently, we noticed the formation of several PVs of varying sizes that fail to form a single acrosome granule in the Golgi phase. We also noticed the formation of more than one acrosome granule nearby several PVs of abnormal size that are scattered around, and in some instances, the acrosome vesicle is not attached to the nuclear envelope. The plausible reason for these abnormalities in acrosome vesicle formation in the early stages of RS development is due to down-regulation of acrosomal-specific genes, such as *Eqtn*, *Tssk1*, *Spaca1/3/4* in KO and KI mice. EQTN1 has been shown to be involved in the trafficking of vesicles to the acrosome vesicle formation site towards the nuclear membrane during acrosome biogenesis [37]. SPACA1 (sperm acrosome-associated-1), an acrosomal membrane protein, participates in the process of acrosome attachment to the nuclear membrane and the disruption of this gene leads to the detachment of the acrosome from the nucleus [38]. As the RS development proceeds to the cap phase in KO and KI mice, we often observed fragmented acrosomal structures with multiple acrosome granules, and the acrosome vesicle fails to grow and expand completely along the nuclear envelope. We believe that pGRTH is involved in the post-transcriptional regulation of some acrosome-vesicle-related genes that are essential for maintaining the structural integrity of the acrosome. These findings are in concert with our previous work showing that pGRTH is required for maintaining the stability of germcell-specific mRNAs until they are ready for translation [19]. In conclusion, our studies highlight the significance of pGRTH in acrosome biogenesis and the progression of round spermatids RS into ES during spermiogenesis.

## 4. Materials and Methods

### 4.1. Animals

All animal experiments were performed in accordance with the guidelines established by the National Institute of Child Health and Human Development Animal Care and Use Committee. Adult wild-type (WT), KO and KI transgenic male mice (8–16 weeks) were maintained at 22 °C in a pathogen-free, light-controlled environment with an alternating light–dark cycle and ad libitum access to water and food.

### 4.2. Isolation of Mice Testicular Cells for scRNA-seq

Testicular cells were prepared individually from three different adult mice (WT, KO and KI) following two-step enzymatic digestion, as described previously [39]. In brief, testes pooled from two mice for each group were dissected and tunica albuginea was removed [40,41]. The seminiferous tubules were treated with collagenase type-I in M199 medium containing 0.1% bovine serum albumin (BSA) for 10 min at 35 °C and dispersed by gently shaking in between. The collagenase-treated tubules were minced and incubated in M199 with 0.1% BSA and 0.1% trypsin for 10 min at 35 °C in rotation at 100 rpm to obtain dispersed cell suspension. The trypsin digestion was stopped by adding 0.02% of trypsin inhibitor (Sigma, St. Louis, MO) to the sample. Cells were filtered through 100 µm and then with 40 µm cell strainer (Millipore, Burlington, MA, USA) to obtain single-cell suspension fraction, pelleted at 800 g for 2 min at 4 °C, washed and resuspended in cold M199 with 0.1% BSA for scRNA-seq.

### 4.3. scRNA-seq Library Preparation, Sequencing and Transcript Counting

scRNA-seq of testicular cells was performed using the 10× Genomics system. In brief, freshly prepared cell suspension (7000 cells for each sample) was loaded into Chromium microfluidic chip used to generate single-cell gel bead emulsions using the 10× Genomics Chromium controller, as per manufacturer’s protocol. Each scRNA-seq library was constructed following the manufacturer’s instructions. Sequencing of libraries was performed for 100 cycle paired-end run on two lanes in SP flow cell using Illumina NovaSeq 6000 instrument (Illumina, San Diego, CA, USA). Raw sequencing data were demultiplexed using the Cell Ranger (10× Genomics) to obtain fastq files for each sample. Alignment of read (mouse mm10), filtering and UMI counting was performed to obtain gene transcript counts per cell (gene barcode matrix) using Cell Ranger count application (10× Genomics).

### 4.4. Cell Type Identification and Clustering Analysis

Cell Ranger filtered count matrices for each sample were imported into R and Seurat R objects (package 4.1.3) were created [42]. For each sample, further filtering was conducted to select high-quality cells. Cells expressing >500 genes and with <10% of transcripts mapped to mitochondrial genes were selected for downstream analysis. Gene expression counts were normalized with a scale factor of 10,000 and a log(1 + n) transformation, using the Seurat NormalizeData function. Seurat objects for the three samples were then combined using SCtransform-based SCTIntegration function. Dimensionality reduction was performed using the top-2000 highly variable genes. Cell clustering and tSNE analysis were performed based on 100 statistically significant principal components. Leiden clustering with resolution 0.3 resulted in 15 clusters. DEGs (marker genes) of each cell cluster were determined by Fold Change threshold above 1.25 using the default Wilcoxon rank-sum test. We then identified cell types (germ cells and somatic cells) using their specific marker gene expression, as previously reported in other scRNA-seq datasets. Waterfall heatmap was generated based on top-20 DEGs for the combined dataset. If a gene turned up to the top DEGs for multiple clusters, they were assigned to the cluster with a maximum fold change.

### 4.5. Pseudotime, Differential Gene Expression and GO Enrichment Analysis

Germ cell clusters (1–12) were used for pseudotime analysis by Monocle 3 R-package [43]. We let the spermatogonia cells (cluster 1) be the root cells and Monocle 3 assigned a pseudotime to each germ cell based on its distance from the root cluster. We ran Monocle3 only once for all three samples together to assign the pseudotime. Comparative trajectory analysis of the three individual samples was performed based on the global pseudotime of the cells in each sample. For differential gene expression analysis on RS cluster between KI vs. WT and KO vs. WT groups, we used the pairwise FindMarkers function of Seurat. Gene ontology functional enrichment analysis was performed on all the DEGs obtained from RS cluster using cluster profiler package in ShinyGO bioinformatics tool.

### 4.6. Isolation of RS from Mice Seminiferous Tubules

RSs were isolated from KO, KI and WT mice testes (10 to 12-week-old), as per the protocol described previously, with minor modifications [44]. Briefly, testes from three individual mice for each group were decapsulated, seminiferous tubules were mildly dispersed and digested using collagenase (1 mg/mL in 1× Krebs buffer; Worthington, NJ, USA) at 37 °C for 3 min to remove Leydig cells. After two washes with Krebs buffer, the tubules were digested with trypsin (0.6 mg/mL in 1× Krebs buffer; Sigma-Aldrich, St. Louis, MO, USA) containing DNase I (ThermoFisher Scientific, Waltham, MA, USA) at 34 °C for 15 min (~15 rpm). The obtained cell suspension was pre-chilled on ice and filtered with 40 μm filter (Millipore). The cells were centrifuged and cell pellet was washed with ice-cold Krebs buffer and mixed with 0.5% BSA, filtered again with 40 μm filter to obtain a single cell suspension. The germ cells (in 0.5% BSA-Krebs Buffer) were loaded onto the prepared BSA gradient (1% to 5% BSA-Krebs Buffer) and allowed to sediment for 90 min on ice. After sedimentation, the cell fractions (1 mL) were collected, washed in ice-cold Krebs buffer and the viability was measured by cell counters (Thermo Scientific). The purity of the RS fractions was verified with DAPI staining (Thermo Scientific) followed by microscopic examination (EVOS M-5000, Thermo Scientific).

### 4.7. Quantitative Real-Time PCR Analysis

Real-time quantitative PCR (qRT-PCR) was used to validate the DEGs obtained from RNA-Seq analysis. Total RNA (0.5 μg) was extracted from RSs isolated from testes of three mice separately for each genotype/group (WT, KO and KI) using Qiagen RNeasy micro kit. Reverse transcription was performed using the SuperScript III first-strand synthesis SuperMix (ThermoFisher Scientific, MA, USA). qRT-PCR experiment was performed with Fast SYBR green master mix in a final volume of 20 μL. All PCR reactions were carried out in triplicate in a 7500 Fast Real-Time PCR machine (Applied Biosystems, Foster City, CA, USA). Cycle threshold (Ct) values were normalized to *Tex29* as reference gene, and relative quantification of transcripts was performed using the comparative 2^−ΔΔCt^ method. The primer sets used in this analysis are available in Appendix A.

### 4.8. Immunohistochemistry

Paraformaldehyde-fixed testicular cross-sections from three individual mice for each group were deparaffinized and rehydrated in a graded series of ethyl alcohol and distilled water. The slides were blocked with 3% normal goat serum in PBS containing 0.05% Tween-20 for 30 min. Then, the slides were incubated overnight with 1:500 dilution of Alexa Fluor 488-conjugated Lectin PNA (Thermo Fisher Scientific) overnight at 4 °C. Subsequently, slides were washed with PBS and mounted using Prolong gold antifade mountant with DAPI. The sections were observed and imaged using EVOS M5000 microscope (Thermo Scientific).

### 4.9. Transmission Electron Microscopy

Adult testis fragments from two individual mice for each group were fixed with 2.5% glutaraldehyde in 0.2 M cacodylate buffer at 4 °C for overnight and then post-fixed with osmium tetraoxide. Thereafter, the samples were en bloc stained with 2% uranyl acetate, dehydrated and embedded in Spurr’s epoxy. Ultra-thin sections were made using Leica EM ultramicrotome and post-stained with lead citrate. Images were taken using JEOL JEM-1400 transmission electron microscope (TEM).

### 4.10. Statistical Analysis

The results are presented as the mean  ±  standard error of the mean (SEM). Data were analyzed using GraphPad Prism 9.0 statistical software (GraphPad Software Inc., San Diego, CA, USA). Differences between groups were determined by two-tailed Student’s *t*-test. *p* < 0.05 was considered statistically significant.

## Figures and Tables

**Figure 1 ijms-24-03127-f001:**
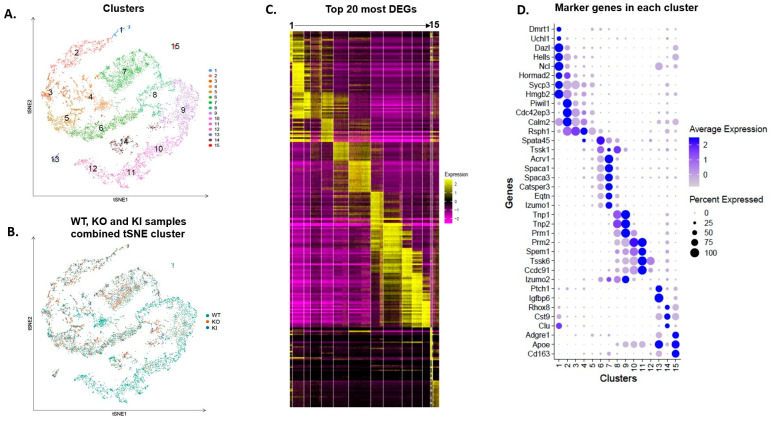
Single-cell transcriptomic profiling of testicular cells from adult WT, KO and KI mice. (**A**) tSNE plot shows profiling/clustering of testicular cells from combined libraries of WT, KO and KI. Each dot represents a single cell and is colored according to its cluster identity. (**B**) tSNE plot of single-cell transcriptome data with cells colored based on their sample/library of origin. (**C**) Heatmaps show the top-20 significantly differentially expressed genes (DEGs) between each cell cluster. (**D**) Dot plot analysis of characteristic marker genes based on cluster ID.

**Figure 2 ijms-24-03127-f002:**
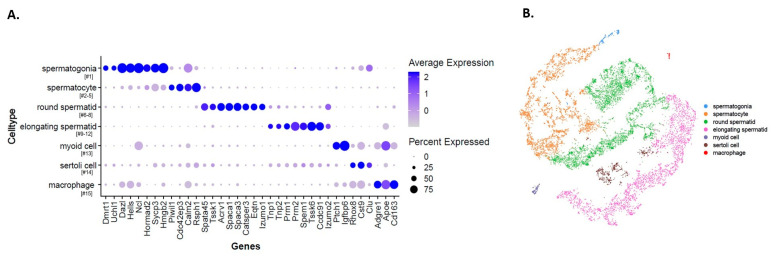
(**A**) Dot plot analysis of characteristic marker genes in different stages of germ cell development during spermatogenesis. (**B**) tSNE representation of spermatogenic and somatic cell-type clusters based on the marker genes identified.

**Figure 3 ijms-24-03127-f003:**
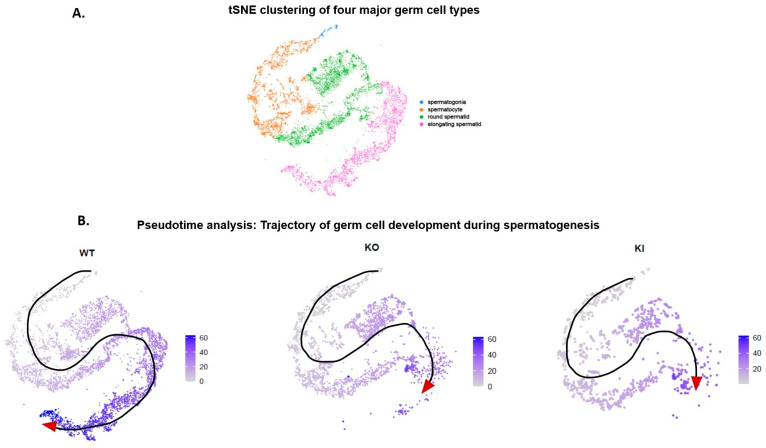
Gene expression dynamics during study state spermatogenesis. (**A**) tSNE plot showing clustering of four major spermatogenic cell types (SG, SP, RS and ES) based on the known marker genes. (**B**) Pseudotime analysis of spermatogenic cell types showing a continuous transition of germ cell development during spermatogenesis process. Trajectory of germ cell development is indicated in arrow.

**Figure 4 ijms-24-03127-f004:**
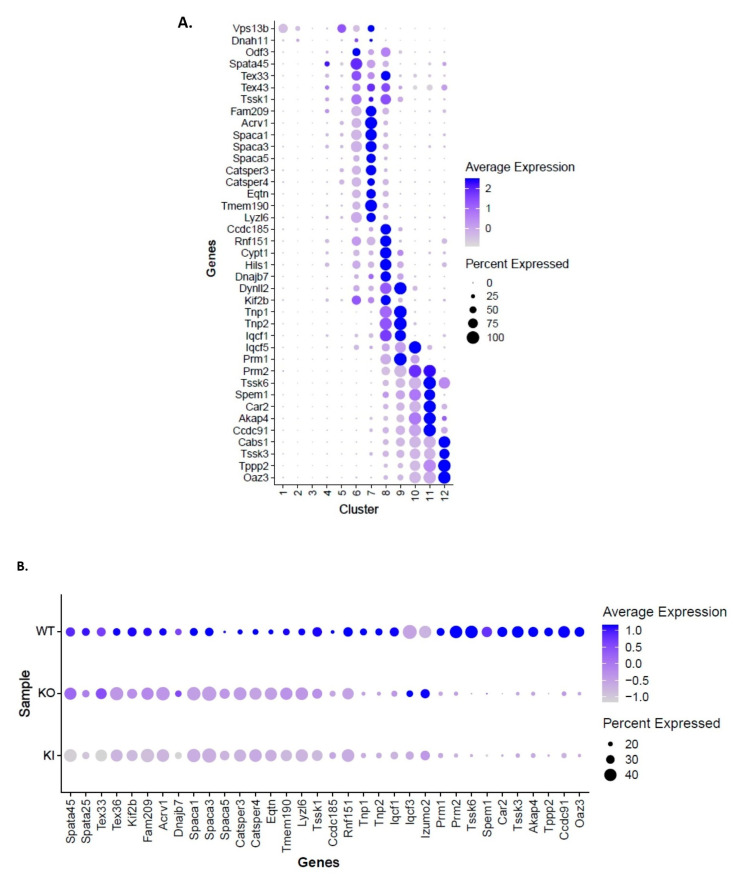
Altered transcriptional profiles in RS clusters of KO and KI during spermiogenesis. (**A**) Dot plot analysis of specific marker genes that are expressed in different stages of RS differentiation transitioning to ESs (condensed spermatids). (**B**) Dot plot showing differential expression of selected marker genes in RS of WT, KO and KI during spermiogenesis. (**C**) GO enrichment analysis on DEGs from RS cluster. Biological process significantly enriched in GO analysis of DEGs from KI to WT and KO to WT groups. (**D**) Cellular components significantly altered in GO analysis of DEGs from KI to WT and KO to WT groups.

**Figure 5 ijms-24-03127-f005:**
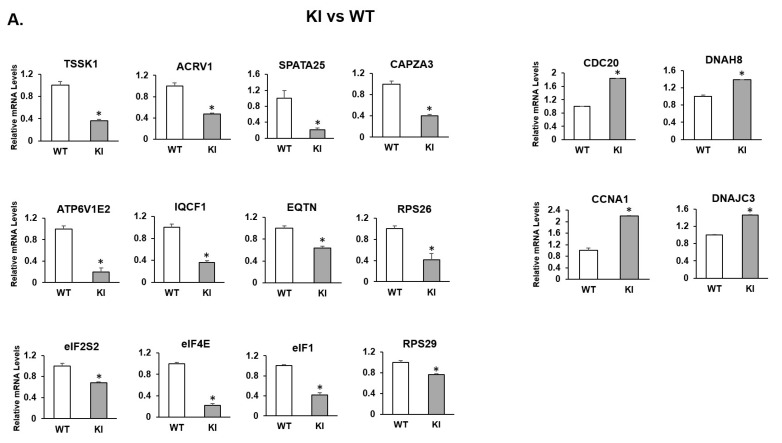
Expression of selected DEGs from RS clusters was measured by qRT-PCR. Bar graphs represent the fold change expression between KI vs. WT (**A**) and KO vs. WT (**B**) groups. Means ± SEM were determined from three independent qRT-PCR experiments with each sample run in triplicate. *p* values were calculated by two-tailed Student’s *t*-test (asterisks indicate *p* < 0.05).

**Figure 6 ijms-24-03127-f006:**
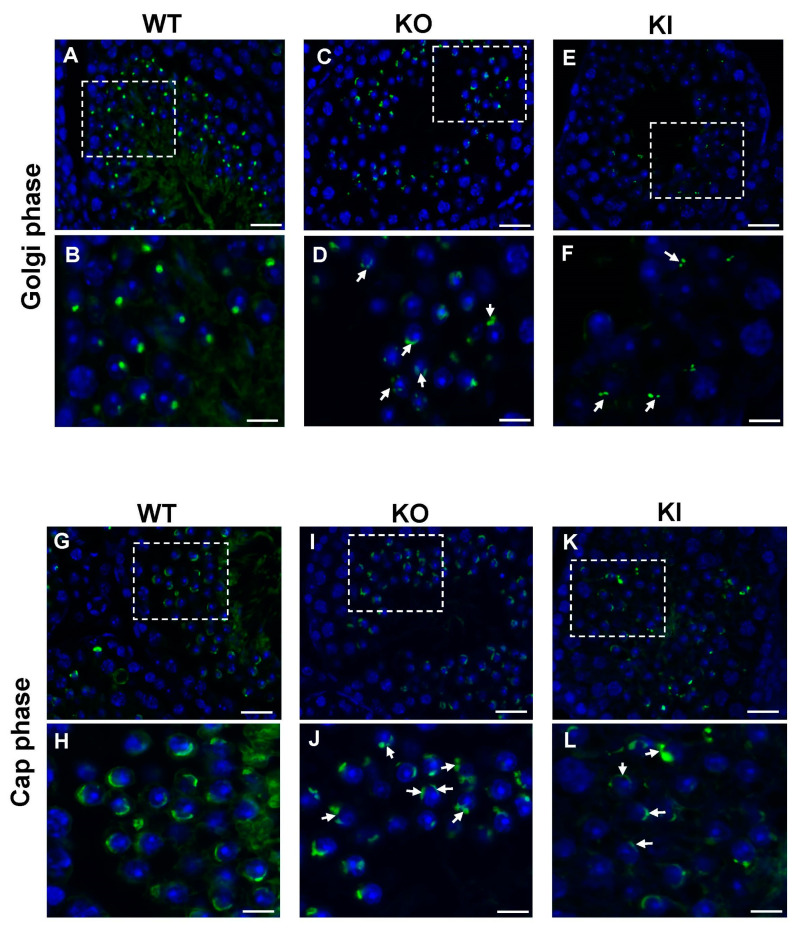
Immunofluorescence detection of developing acrosome in RS of adult mice testis sections using FITC-PNA staining. PNA staining in WT RS in the Golgi phase shows homogenous FITC signal (green) in single acrosome vesicle (**A**,**B**), while RS from both KO and KI mice have a diffuse pattern with more than one acrosome vesicle (**C**–**F**). In the cap phase, WT RSs display the characteristic thin layer of acrosomal caps (**G**,**H**) while KO and KI spermatids have a punctate PNA staining indicating fragmented acrosomal structure (**I**–**L**). DAPI (blue) was used for nuclear staining. Arrows indicate abnormal acrosomes. Lower-magnification images were taken at 60×. Dotted regions in the images were shown at higher magnification taken at 100× oil immersion. Scale bar for low-magnification images is 50 μm and for high-magnification images is 10 μm.

**Figure 7 ijms-24-03127-f007:**
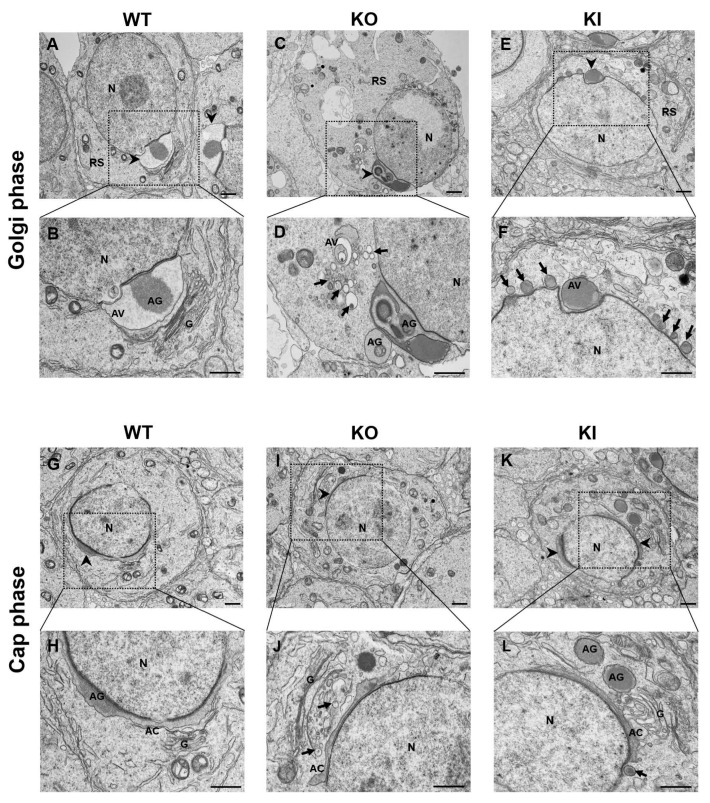
TEM images of developing RS in the testis of KO and KI mice show abnormalities in acrosome formation. WT RSs exhibit the characteristic developmental path of the acrosome. In the Golgi phase of RS of WT (**A**,**B**) proacrosomal vesicles indicated by arrows fuse to create a giant acrosomal granule (AG), which flattens along the acroplaxome and the nuclear envelope exhibiting a cap-shaped acrosomal structure in the cap phase (**G**,**H**). In the Golgi phase of RS of KO and KI, PVs are seen moving toward the nucleus and contacting the nuclear envelope, but they are unable to fuse to form a single AG. Hence, we notice multiple AGs (**C**–**F**). In the cap phase, the acrosome does not exhibit a typical cap-shaped structure and fails to flatten completely in both KO and KI RS. We still observe more than one AG and PV (**I**–**L**). AV: acrosome vesicle, AC: acrosome, AG: acrosome granule, G: Golgi apparatus, N: nucleus, PV: proacrosomal vesicles (indicated by arrows). Arrow heads indicate acrosome vesicle. Dotted regions of the images were shown in higher magnification below. The scale bar for low-magnification images is 1 μm and for high-magnification images is 800 nm.

## Data Availability

The scRNA-seq data were deposited in the GEO portal [https://www.ncbi.nlm.nih.gov/geo/query/acc.cgi?acc=GSE221226]. The accession number for the raw and processed data files is available in GEO: GSE221226.

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
