# Peer review of "Single-Cell Transcriptomic Profiling of the Mouse Testicular Germ Cells Reveals Important Role of Phosphorylated GRTH/DDX25 in Round Spermatid Differentiation and Acrosome Biogenesis during Spermiogenesis"

_ijms, 2023, doi:10.3390/ijms24043127_

Round 1

Reviewer 1 Report

In the article titled “Single Cell Transcriptomic Profiling of Testicular Germ Cells Reveals Important Role of Phosphorylated GRTH/DDX25 in Round Spermatid Differentiation and Acrosome Biogenesis during Spermiogenesis” the authours studied the role of the GRTH in germ cell development at different stages during spermatogenesis, by single cell RNA-seq analysis of testicular cells from adult WT, KO and KI mice and studied dynamic changes in gene expression. They demonstrate the crucial role of pGRTH in differentiation of round spermatid into elongated spermatids, and acrosome biogenesis and its structural integrity.

I thing that this work is interesting but requires a major revision.

My suggestions are as follows:

·       The abstract is too long. Try to stay within 200 words as indicated by the journal or go a little overboard

·       In the introduction you should not put the results so from line 97 to 101 you need to delete and better define the objectives of the study at the end of the introduction

·       The titles of the result paragraphs should not anticipate the result but should indicate the experimental approach and objective at most

·       Improve images and histograms quality

·       I recommend making the histograms in Fig. 5 in different colors for a better understanding of the results. The same color does not make the result well understood. e.g. wt in white, KO in black and KI in gray

·       Indicate the magnification used in fig 6

·       The discussion should not be a summary of the results obtained. It has to be a commentary on the results, you have to more explain the molecular mechanism of action because this journal falls into the molecular biology category, and you have to fit the data obtained with the state of the art.

·       Finally, I would add what external factors may affect the phosphorylation of GRTH/DDX25

Author Response

  • We reduced the abstract to 200 words as per the journal guidelines.
  • As suggested, we removed the summary of results from introduction and explained the objectives of the study clearly at the end. See lines 92-101 in the revised manuscript.

  • As suggested, the subheadings in the results sections were modified to indicate the experimental approach and objective instead of outcome of the results.

  • For initial submission we provided the all images in a single PDF. We are now providing high quality images (300 dpi) for each figure as separate jpg file. 
  • As suggested, we modified the histograms of Fig. 5 using 3 different colors.

  • We have now indicated the magnifications of images in Fig. 6. See line 442.
  • As per your suggestion, we have elaborated the discussion by clearly explaining the biological functions of acrosome and translation related genes that were altered in KI and KO and their relevance to spermatogenesis by explaining their mode of action at molecular level that has been reported by other researchers. To the best of our knowledge, we have mentioned the role of each acrosomal genes in the present study in relation to acrosome formation or function and linked to the acrosomal defects we observed in round spermatids of KI and KO mice. See lines 327-375 and 395-402 in the revised version.

  • In our previously published studies, we have shown that PKA is responsible for the phosphorylation of GRTH [This line was already mentioned in the introduction section in previous version of manuscript]. Inhibitors of PKA and mutations in amino acids (E165/K240/D237) surrounding R242 at the GRTH-PKA interface significantly reduced the phosphorylation of GRTH was reported by our group. We have also shown that cAMP further induces/increases the phosphorylation of GRTH by regulating PKA activity. We think that this information in the discussion section does not fit the context of paper which is about analyzing changes in the transcription profiles of genes using scRNA-seq method. However, we added this information in the introduction section. See lines 84-87 in the revised manuscript.

Reviewer 2 Report

For Author

L.92-101: In present study, ---------------------.

1. The author shows the conclusion in the Introduction. The structure of the article is incorrect. The author should indicate a clear purpose.

2. The disorder of round spermatids to spermatozoa has been reported in KI and KO mice. Why did you perform scRNA-seq again?

Figure 6

3. The author should also show weak magnification.

Figure 7

4. The author should also show weak magnification.

L.262: Discussion

5. Despite the author's analysis of the genes of testicular cells, the content of the discussion is generally poor.

L.302-: In this study, we found the transcripts of acrosomal genes such Acrv1, Tssk1, Tsks, Spata25, Eqtn, Iqcf1 and Atp6ve were significantly downregulated in the round spermatids of KI and KO mice during spermiogenesis.

6. The author did not discuss the altered genes in detail. (Acrv1, Tssk1, Iqcf1, Spaca9, Eqtn, Capza3, Spata25, Eif4e, EiF1, Eif2s2, Rps26, Rsp29, Ccna1, Cdc20, Dnajc3, Dnah8) Why? ?

L.321-: Immunofluorescence detection of acrosomes using FITC-PNA staining revealed several abnormalities throughout the acrosome development during spermiogenesis process. When we carefully examined the ultra-structures of round spermatids using TEM, we found severe defects in acrosome biogenesis in the KO and KI mice from the early stages of spermatid development. 

7. The author should discuss the relationship between altered genes (Up or Down-regulation) and morphology.

L.338: 4. Materials and Methods

8. The number of mice is unknown in all experiments. Why?

L. 446-: Figure S3

9. The author should show in the results.

Author Response

  1. As per the suggestion, we removed the conclusion from the introduction section and provided the objective of our study clearly. See lines 92-101 in the revised manuscript.
  2. In our previous studies, we have shown that in both KO and KI mice the round spermatids fail to elongate, and consequently lack sperm by morphological examination of testis sections. Although previously, we did bulk RNA-seq analysis on the whole testis of KI mice only (KI vs WT), we never performed single cell RNA-seq experiment on both KO and KI mice previously. And we have not determined the precise changes in the gene expression profiles with respect to GRTH role in a specific germ cell type (spermatocytes or round spermatids) in detail. Further we did not perform bulk RNA-seq analysis on KO mice which will allow us to know if there are any differences between KO and KI at whole transcriptome level. Therefore, to understand the precise role of GRTH in different stages of germ cell development (specifically in spermatocytes and round spermatids) we performed scRNA-seq. The scRNA-seq analysis from different germ cell types (meiotic to haploid germ cells) at single cell level will provide a more comprehensive transcriptome landscape which will allow us to better understand the changes in the gene expression profiles at specific stages of germ cell differentiation during spermatogenesis which cannot be revealed using bulk RNA-seq analysis.

  3. We have provided lower magnification images for fig 6.

  4. We have provided lower magnification images for fig 7.

  5. As per your suggestion, we have elaborated the discussion and clearly explained the biological function/role of acrosomal and translation related genes that were altered in KI and KO and their relevance to spermiogenesis in the context of this study by citing the work published by other researchers.

  6. As suggested, we have discussed in detail the functional significance/role of  both down & up-regulated genes [(Acrv1, Tssk1, Iqcf1, Spaca9, Eqtn, Capza3, Spata25, Eif4e, EiF1, Eif2s2, Rps26, Rsp29, Ccna1, Cdc20, Dnajc3, Dnah8] in relation to spermatogenesis process wherever it is necessary and provided additional references to support these findings. See lines 327-375 and 395-402 in the revised manuscript.

  7. To the best of our knowledge, we provided plausible explanation for biological role of most of the altered genes and linked to morphological changes observed in round spermatids of KI and KO mice. See lines 327-375 and 395-402 in the revised manuscript.

  8. We provided number of mice used for all the experimental methods in the revised manuscript. See lines 155, 170, 179 and 187.

  9. We mentioned the Figure S3 in the results. See Lines 270-271.

Reviewer 3 Report

The authors focused on phosphorylated Grth and performed single cell RNA seq analysis using 10x genomics. They compared WT, Grth KO and human GRTH KI with R242Hmutation which lack of pGRTH. The RNA seq analysis showed that significant differential expression of genes critical during spermiogenesis in KO and KI mice and these data are identical with TEM analyses. They performed detailed gene expression analysis. However, I could not fully understand why they used the KI mice (ref. 19).

1.       I could not find human GRTH could rescue the phenotype of Grth deficient mice in reference. They showed the phenotype of KI is almost same as that of Grth deficient mice. If they used human GRTH with R242H mutation KI mice, they should perform human GRTH rescue experiments. Or, they should generate Grth mut mice that have mutation at the identical position of R242.

2.       The TEM data of KI mice in Fig. 7 had been reported in ref. 19.  

Author Response

GRTH-KI transgenic mouse is a valuable model for us to evaluate the role of phospho-GRTH specifically in male germ cells as the R242H mutation leads to loss of phosphorylation of GRTH in mice that are infertile because the spermatogenesis is incomplete. Therefore, to understand the precise role of phospho-GRTH in different stages of germ cell development in a single experiment (specifically in round spermatids during spermiogenesis) we performed scRNA-seq analysis of KI germ cells. The scRNA-seq analysis from different germ cell types (meiotic to haploid germ cells) at single cell level will provide a more comprehensive transcriptome landscape which will allow us to better understand the changes in the gene expression profiles at specific stages of germ cell differentiation during spermatogenesis. Further, we wanted to know if there are any differences between germ cells of KO and KI mice at whole transcriptome level.

  1. In our previous studies, we identified a missense mutation of R242H in GRTH gene in Japanese men with non-obstructive azoospermia. In vitro experiments performed by overexpressing the human mutant GRTH construct in COS-1 cells revealed loss of phospho-GRTH form. Our objective was to find out the phenotypic outcome of mice carrying human GRTH transgene with this mutation in vivo. Interestingly, we found that it’s the phosho-GRTH which plays an important role in spermatogenesis and essential for male fertility as we found similar phenotype to that of GRTH-KO mice. We agree with the reviewer point of view about generating a transgenic mouse with identical mutation (R242H) in the mouse GRTH gene. We will look forward to pursuing this approach in generating such a transgenic mouse for our future work.

  1. Yes, we have previously reported the TEM studies on RS of KI mice showing few abnormalities in acrosome and explained about it superficially. However, we did not perform more detail analysis on the acrosomal defects in different phases of acrosome formation (Golgi phase & cap phase) during round spermatid development and the reason for these defects. In this present study, we looked at the ultrastructure of acrosomes at much higher magnification to better understand the why the acrosome formation is impaired resulting in different abnormalities in both KI and KO mice. We also want to see if similar acrosomal defects will be detected in KO mice as we did not perform these TEM studies on RS of KO mice at higher magnifications. Moreover, changes in transcription profiles of specific acrosomal genes observed in RS by scRNA-seq data warranted these TEM studies.

Round 2

Reviewer 1 Report

Accept in present form

Author Response

We carefully performed spell checks through the manuscript and rectified minor grammatical errors.

Reviewer 2 Report

For Author

The authors answered the reviewers' questions correctly except for question 8. However, Reviewer think the number of mice in the scRNA-seq experiment is statistically small. If the number of mice is correct, please cite some references.

Author Response

There are published articles on scRNA-seq on testicular germ cells in reputed journals where researchers have performed scRNA-seq analysis using one or two mice. For example, Grive et al., work on scRNA-seq analysis on postnatal testis was performed using single mice for each sample group [PLOS Genetics, 2019: e1007810. https://doi.org/ 10.1371/journal.pgen.1007810]. An article “Single-cell RNA sequencing of adult mouse testes” published in Nature Science Data journal in 2018, authors used testes from two mice. We have provided above mentioned two references in the line 111, and references were included. In addition, the libraries for each sample in out study were run in two independent lanes using SP flowcell in Illumina Novaseq 6000 machine to obtain more number of reads. This is now mentioned in the line 124 in the revised manuscript.

For scRNA-seq the robustness of data depends on number of cells (at least 1000-2000 cells) obtained and average mapped reads (20-30K reads/cell) for each sample.

Reviewer 3 Report

I thought present data could not indicate phosphorylated GRTH is essential for round spermatid differentiation and acrosome formation. I agree that previous reports showed R242H mutation found in Japanese men with non-obstructive azoospermia. However, I think they should demonstrate human GRTH play the same role of mouse GRTH in mouse testis or mouse GRTH with a mutation in the identical position of R242H show the phenotype of abnormal spermatogenesis.

Author Response

Although, we agree with the reviewer that transgenic with R242H mutation in the mouse GRTH gene will be useful to show the phenotype of abnormal spermatogenesis in mice. There is 93% similarity between human and mouse GRTH protein amino acid sequence and structurally is equally close. Also, the amino acid sequence [KLIDLTKIRVFVLDEAD] at the phosphorylation site Threonine T239 and structurally close the mutated R242 site is 100% similar for both mouse and human GRTH. For these reasons, we are certain that if we generate a GRTH transgenic mice with R242H mutation in mouse gene we will obtain the similar phenotype with spermatogenic arrest leading to male infertility.

For the present study, it is not required to generate/create a new homozygous GRTH transgenic mice with R242H mutation in mouse GRTH gene for the reasons that I explained above. In this study, we are addressing our existing well characterized KO and KI transgenic models we have developed through the years of thorough examination. Also, we estimate it will take at least 6 months to make this new transgenic mouse line requested and additional couple of more months to perform experiments on a new homozygous mice line with a specific age group (8-10 weeks) to study the phenotype to see morphological abnormalities. Therefore, we believe that it is not appropriate to include this new transgenic mouse work in this study.